# Ionic Liquid Extraction Behavior of Cr(VI) Absorbed on Humic Acid–Vermiculite

**DOI:** 10.3390/molecules26247478

**Published:** 2021-12-10

**Authors:** Hsin-Liang Huang, P.C. Lin, H.T. Wang, Hsin-Hung Huang, Chao-Ho Wu

**Affiliations:** Department of Safety, Health and Environmental Engineering, National United University, Miaoli 36063, Taiwan; sandy23063051@hotmail.com (P.C.L.); flower611003@yahoo.com.tw (H.T.W.); hsinhunghuang@gmail.com (H.-H.H.); a445dd@gmail.com (C.-H.W.)

**Keywords:** hexavalent chromium, humic acid, vermiculite, ionic liquid, NMR

## Abstract

Cr(VI) can be released into soil as a result of mining, electroplating, and smelting operations. Due to the high toxicity of Cr(VI), its removal is necessary in order to protect ecosystems. Vermiculite is applied in situations where there is a high degree of metal pollution, as it is helpful during the remediation process due to its high cation exchange capacity. The Cr(VI) contained in the vermiculite should be extracted in order to recover it and to reduce the impact on the environment. In this work, adsorption equilibrium data for Cr(VI) in a simulated sorbent for soil remediation (a mixture that included both humic acid (HA) and vermiculite) were a good fit with the Langmuir isotherm model. The simulated sorbent for soil remediation was a favorable sorbent for Cr(VI) when it was in the test soil. An ionic liquid, [C_4_mim]Cl (1-butyl-3-methylimidazolium chloride), was studied to determine its efficiency in extracting Cr(VI) from the Cr- contaminated simulated sorbent in soil remediation. At 298 K and within 30 min, approximately 33.48 ± 0.79% of Cr(VI) in the simulated sorbent in soil remediation was extracted into [C_4_mim]Cl. Using FTIR spectroscopy, the absorbance intensities of the bands at 1032 and 1010 cm^−1^, which were attributed to C-O bond stretching in the polysaccharides of HA, were used to detect the changes in HA in the Cr-contaminated simulated sorbent for soil remediation before and after extraction. The results showed that Cr(VI) that has been absorbed on HA can be extracted into [C_4_mim]Cl. Using ^1^H NMR, it was observed that the 1-methylimizadole of [C_4_mim] Cl played an important role in the extraction of Cr(VI), which bonded with HA on vermiculite and was able to be transformed into the [C_4_mim]Cl phase.

## 1. Introduction

Arsenic, cadmium, chromium, mercury, nickel, lead, zinc, and copper are the major metals that are often found in contaminated soil. According to studies conducted by the Taiwan Environmental Protection Agency (EPA), thirteen out of sixty-two remediation sites are presently contaminated with metals: mainly nickel, chromium, lead, and zinc, followed by copper and cadmium [1].

Cr is naturally present in its oxidized state of Cr(III), whereas divalent, tetravalent, and pentavalent Cr are unstable in the environment. Cr(III) is one of the essential elements involved in protein metabolism in animals and humans [2]. Cr(VI) is genetically toxic to cells; it has been shown to be a carcinogen as it affects the functions of deoxyribonucleic acid [3]. It is possible for Cr(VI) to be absorbed into edible plants or vegetables, resulting in them having reduced root and coleoptile growth [4,5]. Moreover, the Cr(VI) concentration in plants depends on the soluble fraction of it that is present in the soil [6,7]. Once it has entered the food chain, Cr(VI) may cause harm to both animal and human organisms when they ingest these affected edible plants or vegetables. Animal studies by the Institute of Labor, Occupational Safety and Health, Taiwan Ministry of Labor have shown that Cr(VI) can cause malignant tumors, while Cr(III) does not [8]. The United States EPA has classified Cr(VI) as a Group A carcinogen in humans, while Cr(III) is classified as a Group D unclassified substance. Therefore, the removal of Cr(VI) from contaminated soils is essential to avoid its toxic impact on ecosystems. High concentrations of Cr(VI) are often found to bind to humic acid (HA), vermiculite, and kaolin clay [9]. Vermiculite, which has a high cation exchange capacity, is a group of hydrated laminar minerals. It is often used as an additive to improve soil structure. Vermiculite is also used as a sorbent in order to absorb the metals that are present in contaminated soil. This results in the accumulation of high concentrations of chromium in vermiculite [10,11]. It has also been reported that Cr(VI) can be absorbed by HA. As the pH increases from 1 to 7, the Cr(VI) adsorption to HA decreases from 100% to 34% [12]. Moreover, the efficiency of Cr(VI) removal increases at higher temperatures [13]. In addition, in one study using peanut shells as a sorbent, the Cr(VI) removal ratios increased from 50% to 90% in an aqueous solution when the reaction time was increased from one to five hours [14]. Remediation technologies for soil that has been contaminated with metals include biological methods, soil washing, solidification/stabilization, and extraction [15,16,17]. Soil washing is one of the most commonly used techniques that can be applied for the remediation of metal-contaminated soils. Cleaning agents, such as surfactants, are used to remove metals from contaminated soil by extracting these toxins into a liquid phase [18]. Soil washing can also be applied for the recovery of the metals that were used in the sorbents for soil remediation.

Ionic liquids (ILs) have special chemical and physical properties, including high thermal stability, negligible vapor pressure, a broad liquid phase range, and excellent electric conductivity [19]. ILs can be used to replace conventional organic solvents and their impact on the environment is considered to be minimal; therefore, they have also been referred to as green solvents [20]. ILs are made of various ions, thereby possessing the properties of salts. Varying the compositions of anions and cations results in different ILs that can be used for different applications. ILs that remain in a liquid phase at room temperature are called room temperature ILs (RTILs) [21]. Kozonoi et al. utilized 1-butyl-3-methylimidazolium nonafluorobutanesulfonate ([bmi][NfO]) to extract Cs^+^, Na^+^, Li^+^, Sr^2+^, Ca^2+^, and La^3+^ ions from aqueous solutions, with extraction ratios of 5, 24, 39, 79, 81, and 98%, respectively [22]. Metal ions with a higher valence can be more easily extracted with ILs. In addition, the extraction efficiencies of metal ions, such as copper, lead, and sodium, are greater when ILs are used than those are achieved with regular organic solvents, e.g., chloroform [23].

The structure of the Cr(VI) complexes that have been formed in the sorbent for remediation in contaminated soil are too complex to reveal the mechanism of absorbance phenomena. In order to understand the effects of Cr(VI) that has been absorbed on vermiculite and that has been extracted with the ionic liquid, a mixture comprising both humic acid (HA) and vermiculite was prepared to simulate the sorbent for remediation in a contaminated soil sample. The adsorption equilibrium data for Cr(VI) in the simulated sorbent were established. During the extraction process, 1-butyl-3-methylimidazolium chloride ([C_4_min]Cl) was used to extract Cr(VI) from the Cr-contaminated simulated sorbent for soil remediation, and the extraction mechanism was explored.

## 2. Results and Discussion

### 2.1. Adsorption of Chromium Species on the Simulated Sorbent in Soil

Table 1 lists the absorption efficiencies of Cr(VI) to the HA, vermiculite, and the simulated sorbent for soil remediation. As shown in Table 1, HA had a high Cr(VI) absorption capacity. The absorption efficiency of Cr(VI) onto HA was greater than that of Cr(VI) onto vermiculite (81.32 ± 1.05% vs. 64.47 ± 1.62%). The absorption efficiency of Cr(VI) onto the simulated sorbent for soil remediation that included both HA and vermiculite was 91.78 ± 1.82%, showing that HA and vermiculite had a synergistic effect on the absorption efficiency of Cr(VI) into the sorbent for soil remediation.

The speciation of chromium on the simulated sorbent in soil remediation was studied using XANES spectroscopy (see Figure 1). The pre-edge intensity of the *3d* elements with *T_d_* symmetry was greater than those with *O_h_* symmetry. The intense peaks for the tetrahedral species of the *3d* transition metals in the pre-edge range were attributable to the *p* component in the *d-p* hybridized orbital. The number of *d*-electrons that the tetrahedral species has affects the intensity of the pre-edge peak [24,25]. The existence of Cr(VI) and Cr(III) in the simulated sorbent in soil remediation was observed by the pre-edge feature that was centered at 5993–5994 eV in the XANES spectra and are distinctive of the deconvolution that takes place during component fitting. In the simulated sorbent in soil remediation, Cr(VI)-HA and Cr(VI)_ads_ were the main species that were present, as seen in Figure 1. It was also determined that about 11% of the Cr(VI) compound was reduced to Cr(III). Cr(VI) can interact with the carboxyl groups of HA, resulting in the reduction of Cr(VI) [26,27].

### 2.2. Adsorption Equilibrium of Chromium Species onto Simulated Sorbent for Soil Remediation

In the adsorption experiments, the concentration of the adsorbed Cr(VI) that was present in the simulated sorbent in soil increased as *C_e_* increased (see Figure 2). In Table 2, the model parameters for Cr(VI) absorption onto the simulated sorbent for soil remediation were estimated by fitting the experimental data. According to the values of the correlation coefficient (*R*^2^), the Cr(VI) absorption onto the simulated sorbent for soil remediation tended to follow the Langmuir and Freundlich isotherm equations in the *C*_0_ range of 1000–8000 mg/L. The Langmuir isotherm fit the experimental data better than the Freundlich isotherm. In the Langmuir isotherm model, the monolayer saturation capacity of the Cr(VI) that was present in the simulated sorbent in soil remediation was 5.57 mg/g. The calculated *R_L_* value from the Langmuir isotherm model was 0.139, which indicated favorable Cr(VI) absorption onto the simulated sorbent for soil remediation. Moreover, the value of 1/*n* in the Freundlich isotherm model was 0.325, which also indicated that Cr(VI) was favorably absorbed on the simulated sorbent for soil remediation.

### 2.3. Extraction of Chromium Species from Cr-Contaminated Simulated Sorbent for Soil Remediation with [C_4_mim]Cl

The compound [C_4_mim]Cl, a hydrophilic IL, can be used to extract HA and is able to intermix with Cr(VI) [28,29]. Thus, in this study, [C_4_mim]Cl was used to extract Cr(VI) from the Cr-simulated sorbent in soil remediation. The results show that approximately 33.48 ± 0.79% of the Cr(VI) was extracted into [C_4_mim]Cl (see Table 3). To understand whether different matrices affected the extraction efficiency of Cr(VI), HA and vermiculite were also tested. The results show that the extraction efficiencies of Cr(VI) compared to those of HA and vermiculite were about 82.85 ± 0.96 and 21.97 ± 1.11%, respectively, showing that HA and vermiculite affected the extraction efficiency. In a similar study, approximately 70% of Cr(VI) that had been chelated with HA in a mesoporous sorbent was able to be extracted into [C_4_mim]Cl [30]. Therefore, the extraction efficiency of Cr(VI) in HA was higher than that in the vermiculite and in the simulated sorbent for soil remediation. 

### 2.4. FTIR Analysis

To further understand the extraction mechanism of [C_4_mim]Cl, the structures of different matrices were tested both before and after extraction using FTIR spectroscopy. As shown in Figure 3, the band at 1088 cm^−1^ was attributed to the C-O group stretching in the ester. The two bands found at 1032 and 1010 cm^−1^ corresponded to the C-O group stretching in polysaccharides [31]. In Figure 3a,b, the broadened peak at 1088 cm^−1^ and the red shifts (ν, cm^–1^: 1032→1011 and 1010→960) were found because of the interaction between HA and vermiculite in the simulated sorbent for soil remediation. Moreover, the bands at 1011 and 960 cm^−1^ were also shifted to 999 and 958 cm^−1^, respectively, in the Cr-simulated sorbent for soil remediation (see Figure 3b,c). It was clear that the Cr(VI) had been absorbed onto the C-O bonds of HA interacted with vermiculite. In Figure 3d, a band shift (999→1001 cm^−1^) was identified when [C_4_mim]Cl was used to extract Cr(VI) from the Cr-contaminated simulated sorbent for soil remediation, showing that the Cr(VI) that had been absorbed on HA was able to be extracted into [C_4_mim]Cl. Furthermore, the changes in the peak at 1088 cm^−1^ were barely observable during extraction with [C_4_mim]Cl (see Figure 3b–d). A slight perturbation was observed in the vermiculite during extraction with [C_4_mim]Cl.

### 2.5. ^1^H NMR

The ^1^H NMR spectra of [C_4_mim]Cl were also measured (see Figure 4). The yield of [C_4_mim]Cl was 98.2%. The structure and impurities of [C_4_mim]Cl are shown in Figure 4a. The ^1^H NMR analysis of [C_4_mim]Cl revealed values of δ 9.66 (s, 1 H), 8.00 (t, *J* = 1.6 Hz, 1 H), 7.87 (t, *J* = 1.6 Hz, 1 H), 4.23 (t, *J* = 7.2 Hz, 2 H), 3.09 (s, 3 H), 1.75 (m, 2 H), 1.22 (m, 2 H) and 0.84 (t, *J* = 7.6 Hz). In Figure 4b, interactions occurred between [C_4_mim]^+^ and the different extracts. Therefore, a downshift of the protons in the imidazole ring (δ 9.66→9.69) was observed. The chemical structure of the imidazole ring in [C_4_mim]^+^ was slightly disturbed in the presence of vermiculite (see Figure 4c). However, HA and Cr(VI) were the main species that were observed to interact with the imidazole ring in [C_4_mim]^+^ because the same field shifts (δ 9.66→9.68 and 9.66→9.72) were also obtained in Figure 4d,e. Note that less 1% of the Cr(III) compounds was able to be extracted into [C_4_mim]Cl [32]. Moreover, the intensities at δ 3.69 were diminished due to interactions between the methyl protons in [C_4_mim]^+^ and the extracts (vermiculite, humic acid, and Cr(VI)), as seen in Figure 4. The 1-methylimidazole in [C_4_mim]Cl played an important role in extracting the Cr(VI), which bonded with HA on the vermiculite, transforming it into the [C_4_mim]Cl phase. Weaker interaction between [C_4_mim]Cl and the vermiculite was shown to affect the extraction efficiency of Cr(VI).

## 3. Materials and Methods

### 3.1. Preparation of Simulated Sorbent for Soil Remediation and Cr-Contaminated Simulated Sorbent for Soil Remediation

In order to synthesize the simulated sorbent for soil remediation, 1.5 g of HA (humic acid sodium salt, Sigma-Aldrich, St. Louis, MO, USA), 6 g of vermiculite (Aldrich, St. Louis, MO, USA), and 50 mL of H_2_O were stirred in a 150 mL beaker for 1 d, filtered, dried at 343 K, and ground. To prepare the Cr(VI)-contaminated HA, vermiculite, and simulated sorbent for soil remediation, 7.5 g of either HA, vermiculite, or simulated soil were incubated with 10 mL of 1000 mg/L of the Cr(VI) solution for 1 h at 298 K. The 1000 mg/L Cr(VI) solution was prepared from 0.25 g of K_2_Cr_2_O_7_ (99%, Sigma-Aldrich, St. Louis, MO, USA) in 250 mL of H_2_O at 298 K. To calculate the adsorption efficiencies, the adsorbed Cr(VI) on humic acid, vermiculite, and simulated sorbent for soil remediation was digested and the chromium concentrations were measured by AA (Hitachi Z-5000, Hitachi Instruments Co., Tokyo, Japan).

### 3.2. Adsorption Isotherm

For the adsorption isotherm experiments, 2000, 3000, 4000, 5000, 6000, 7000, and 8000 mg/L Cr(VI) solutions were prepared using methods that were similar to those used for the preparation of the 1000 mg/L Cr(VI) solution was. Samples of 10 mL of each of these different concentration solutions was mixed with 7.5 g of each simulated sorbent for soil remediation in test tubes; the test tubes were then shaken at 298 K for 4 h. All of the experiments were run in five replicates.

The Langmuir and Freundlich adsorption equations were used to explain the adsorption isotherms:

Langmuir model:*q*_*e*_ = (*q*_*m*_*K*_*L*_*C*_*e*_)/(1 + *K*_*L*_*C*_*e*_)
where *q_e_* (mg/g) is the equilibrium concentration of the absorbed chromium in the simulated sorbent for soil remediation, *C_e_* (mg/L) is the equilibrium concentration of the chromium in the solution, and *K_L_* (1/mg) and *q_m_* (mg/g) are the constants.
*R_L_* = 1/(1 + *K_L_C_o_*)
where *R_L_* is the equilibrium parameter, and *C_o_* is the initial chromium concentration in the solution (mg/L). The value of *R_L_* suggests the tendency of the isotherm to be irreversible (*R_L_* = 0), favorable (0 < *R_L_* < 1), linear (*R_L_* = 1), or unfavorable (*R_L_* > 1).

Freundlich model:*q_e_* = *K_F_**C_e_*^1/*n*^
where *q_e_* (mg/g) is the equilibrium concentration of the absorbed chromium in the simulated sorbent for soil remediation, *C_e_* (mg/L) is the equilibrium concentration of the chromium in the solution, and *K_F_* and *n* are the constants that are associated with the adsorption capacity ((mg/g)(L/mg)^1/*n*^) and the adsorption intensity, respectively.

### 3.3. Synthesis of [C_4_mim]Cl

To prepare [C_4_mim]Cl, equal numbers of moles of 1-methylimidazole (99%, Sigma-Aldrich, St. Louis, MO, USA) and 1-chlorobutane (99%, Alfa-Aesar, Kendal., Germany) were mixed, stirred, and refluxed in a 250 mL flask at 343—353 K for 4 d. After cooling, 30 mL of ethyl acetate (99.9%, J.T. Baker, Phillipsburg, NJ, USA) was added to remove the unreacted 1-methylimidazole. Moreover, any unreacted matter was removed from the [C_4_mim]Cl by means of a rotary evaporator (N-1300VF, EYELA, Tokyo, Japan).

### 3.4. Extraction of Cr(VI) from Sorbents with [C_4_mim]Cl

In order to test the extraction efficiency of the Cr(VI) by [C_4_mim]Cl, 0.4 g of Cr(VI) -contaminated HA, vermiculite, and the simulated sorbent for soil remediation were incubated with 1.5 g of [C_4_mim]Cl in 0.5 mL of H_2_O in glass tubes that were each shaken for 30 min. Afterwards, the mixture was filtered to separate the [C_4_mim]Cl from the solids. The solids were washed with deionized water several times to remove any residual [C_4_mim]Cl that was present in the HA, vermiculite, and the simulated sorbent for soil remediation. The resulting [C_4_mim]Cl was removed using H_2_O and a rotary evaporator. In order to clarify the influence of [C_4_mim]Cl on the extraction process, H_2_O was used as the extraction solution without [C_4_mim]Cl using similar extraction steps. All of the experiments were run with five replicates. Moreover, a blank experiment was also carried out. The extraction efficiencies of Cr(VI) from the Cr(VI)-contaminated HA, vermiculite, and the simulated sorbent for soil remediation into [C_4_mim]Cl were estimated by the following equation:Extraction efficiency (%) = (*C*_1_ − *C*_2_)/*C*_1_ × 100%
where *C*_1_ is the initial chromium concentration in the humic acid, vermiculite, and the simulated sorbent for soil remediation (mg/g), and *C*_2_ is the chromium concentration in the humic acid, vermiculite, and the simulated sorbent for soil remediation after extraction with [C_4_mim]Cl (mg/g).

### 3.5. Concentrations of Cr(VI) Absorbed on Sorbents and Extracted into [C_4_mim]Cl

Samples of 0.5 g of Cr(VI)-contaminated HA, vermiculite, or simulated sorbent for soil remediation and the resulting [C_4_mim]Cl were each digested in 6 mL of HCl (37%, Riedel-de Haën, Seelze, Germany) and 2 mL of HNO_3_ (65%, Merck, Darmstad, Germany) using a microwave digestion system (CEM MARS 6, Mattews, NC, USA). The digestion temperature was increased from room temperature to 448 K by means of a 10 K/min heating rate that took place over 10 min [33]. The final total volume was made up to 50 mL. An AA spectrometer was applied to determine the chromium concentrations in the Cr(VI)-contaminated HA, vermiculite, or the simulated sorbent for soil remediation and the resulting [C_4_mim]Cl. The Cr concentration of the calibration curves ranged from 0.1–5.0 mg/L, with a correlation coefficient > 0.9995. Each sample was measured three times, and the mean was calculated automatically using AA.

### 3.6. Spectroscopic Analysis

The Cr K-edge XAS (X-ray absorption spectroscopy, 16A1, National Synchrotron Radiation Research Center, Hsinchu, Taiwan) spectra of the chromium on the simulated sorbent for soil remediation was recorded on the Wiggler beam line (16A1) at the Taiwan National Synchrotron Radiation Research Center. The electron storage ring was operated at an energy of 1.5 GeV and at a current of 300 mA. A chromium foil absorption edge at 5989 eV was used to calibrate the photon energy. To measure the Cr K-edge absorption spectra, the fluorescence mode on a Lytle detector was used. The XANES (X-ray absorption near edge structure) spectra of chromium model compounds, such as CrCl_3_·6H_2_O, K_2_CrO_7_, Cr(NO_3_)_3_, Cr(OH)_3_, Na_2_CrO_4_, Cr_2_O_3_, CrO_3_, Cr(VI)-HA, Cr(III)-HA, Cr(VI)_ads_ (by impregnation of K_2_CrO_7_ (3 wt%) on vermiculite), Cr(VI)_ads_ (by impregnation of CrCl_3_·6H_2_O (3 wt%) on vermiculite), Cr(VI) ion (prepared by dissolution of 0.5 g K_2_CrO_7_ in 50 mL H_2_O), Cr(III) ion (prepared by dissolution of 0.5 g CrCl_3_·6H_2_O in 50 mL H_2_O), and Cr foil were also measured. Cr(VI)-HA was prepared by mixing 0.5 g of K_2_CrO_7_ and HA in 50 mL of deionized water, and then filtering and drying the mixture at 343 K. To prepare Cr(VI)-HA and Cr(III)-HA, 7.5 g of HA was incubated with 10 mL of 1000 mg/L Cr(VI) and Cr(III) solution, respectively, for 1 h at 298 K, and the mixture was then filtered and dried at 343 K.

The solid samples and KBr (99.5%, Panreac, Barcelona, Spain) were uniformly mixed and ground at a ratio of 1 to 100. The mixture was pulverized in an agate mortar, and the mixed dyes were prepared in quantities of 5–8 tons. An FTIR spectrometer (Thermo Nicolet 6700, Thermo Fisher Scientific, Waltham, MA, USA) was used to investigate the structures of the HA, vermiculite, and Cr-simulated sorbent from soil remediation before and after extraction in the wavenumber range of 4000–400 cm^−1^, with 32 scans, and a 4 cm^−1^ resolution. The ^1^H NMR spectra of the [C_4_mim]Cl were also determined on a Bruker Avance 300 spectrometer (Bruker, Billerica, MA, USA) with tetramethyl silane (TSM) as an internal standard (acquisition time = 1.373 s, actual pulse repetition time = 2 s, number of scans = 32, and excitation pulse angle = 30°).

## 4. Conclusions

The absorption efficiencies of Cr(VI) onto HA, vermiculite, and the simulated sorbent for soil remediation were 81.32 ± 1.05%, 64.47 ± 1.62%, and 91.78 ± 1.82%, respectively. The simulated sorbent for soil remediation that contained HA and vermiculite had a higher Cr(VI) adsorption efficiency. The experimental absorption data were fitted better by the Langmuir isotherm model than by the Freundlich isotherm model. It was also found that Cr(VI) was favorably absorbed onto the simulated sorbent for soil remediation. During the extraction of Cr(VI) from the Cr-contaminated simulated sorbent for soil remediation with [C_4_mim]Cl, about 33.48 ± 0.79% of the Cr(VI) was extracted into [C_4_mim]Cl. The results of the FTIR spectra showed that Cr(VI) could adsorb onto HA, which interacted with vermiculite in the simulated sorbent for soil remediation. After extraction, the absorbed Cr(VI) could be extracted from the simulated sorbent for soil remediation into [C_4_mim]Cl. An interaction between the 1-methylimidazole of the [C_4_mim]Cl and Cr(VI) was also observed during extraction by ^1^H NMR. The [C_4_mim]Cl and vermiculite had weaker interaction, which effected the Cr(VI) extraction efficiency. This work exemplifies that the ^1^H NMR technique can reveal the changes that take place in [C_4_mim]Cl during extraction of chromium species from Cr-contaminated simulated sorbent into [C_4_mim]Cl.

## Figures and Tables

**Figure 1 molecules-26-07478-f001:**
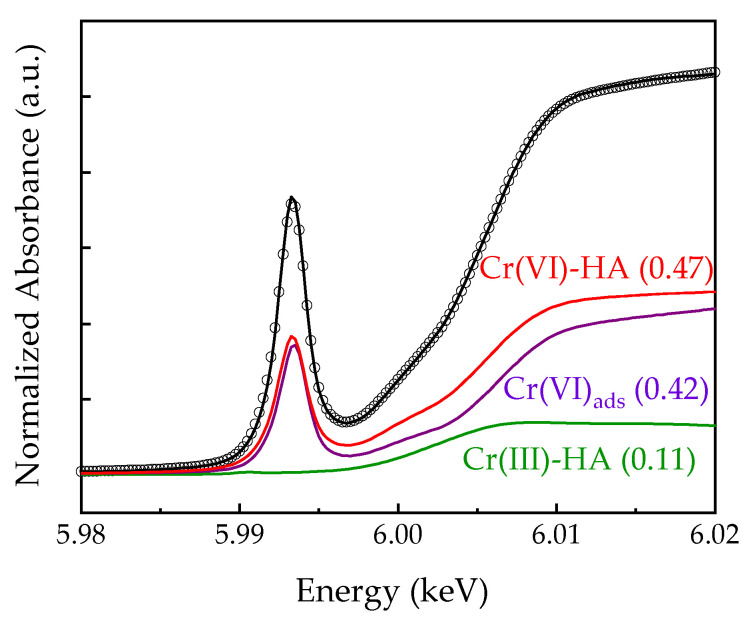
The least-square fitted XANES spectra of chromium in HA–vermiculite.

**Figure 2 molecules-26-07478-f002:**
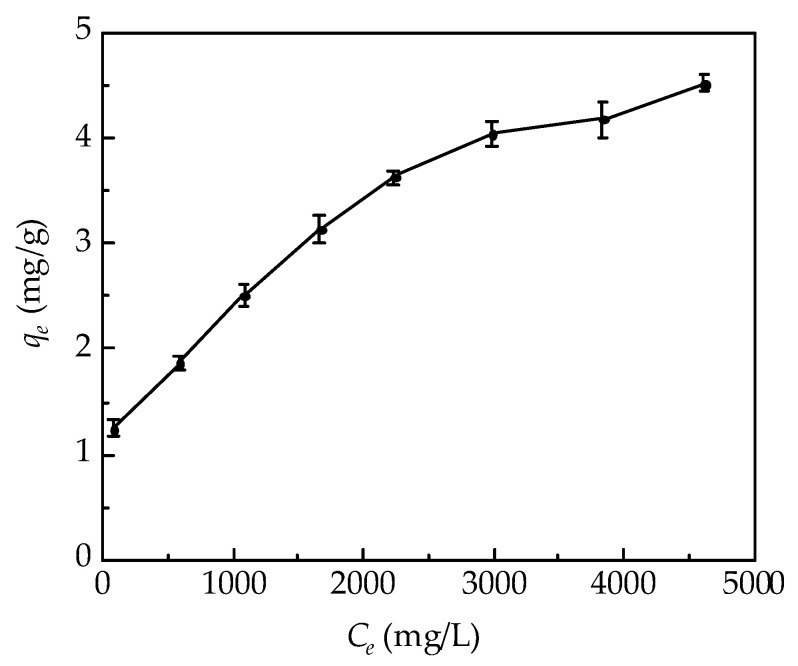
Experimental adsorption equilibrium results for the absorption of Cr(VI) onto the simulated sorbent for soil remediation. Error bars show the standard deviation of five replicates.

**Figure 3 molecules-26-07478-f003:**
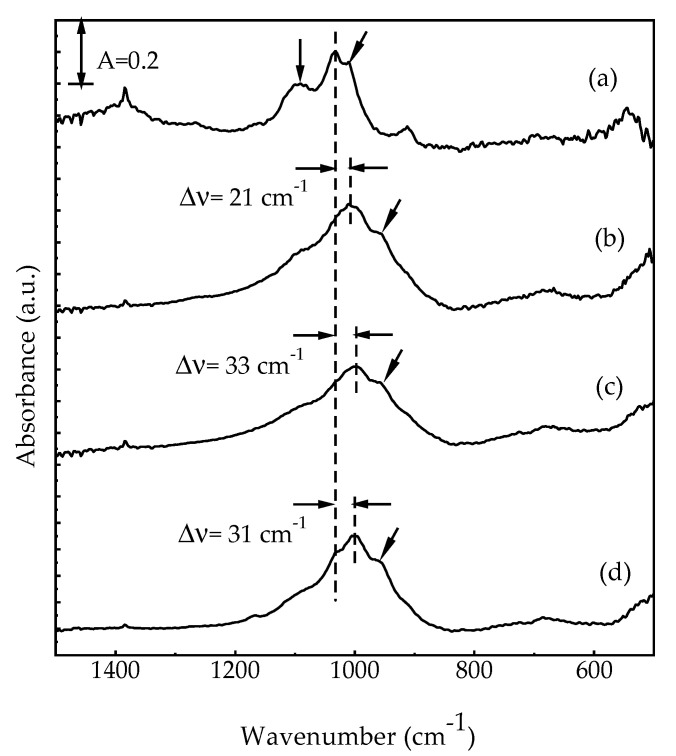
Fourier-transform infrared spectra of (**a**) humic acid, (**b**) simulated sorbent for soil remediation, and Cr–contamined simulated sorbent for soil remediation (**c**) before and (**d**) after extraction with [C_4_mim]Cl.

**Figure 4 molecules-26-07478-f004:**
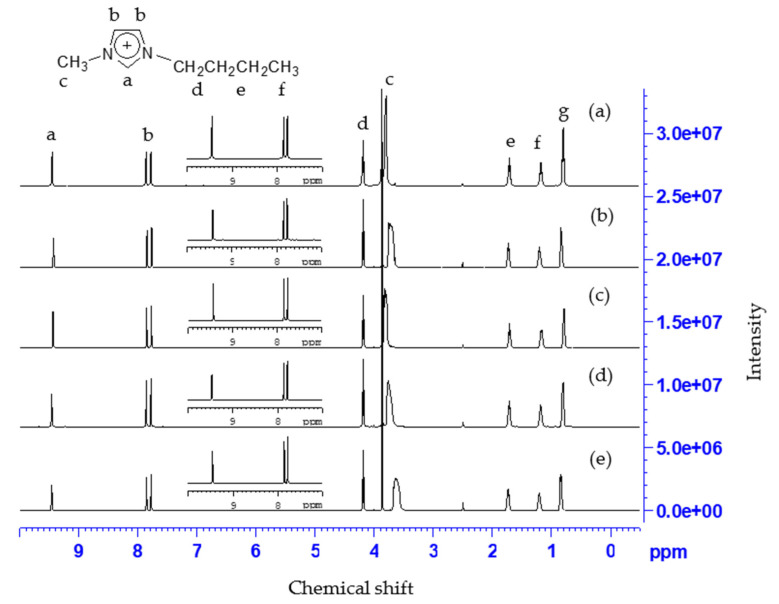
The ^1^H NMR spectra of (**a**) [C_4_mim]Cl, (**b**) Cr-contaminated simulated sorbent in soil remediation–extracted [C_4_mim]Cl, (**c**) vermiculite-extracted [C_4_mim]Cl, (**d**) humic acid–extracted [C_4_mim]Cl, and **(e)** Cr(VI)–extracted [C_4_mim]Cl.

**Table 1 molecules-26-07478-t001:** Absorption efficiencies of Cr(VI) onto sorbents.

Absorbed Cr(VI) on:	Absorption Efficiency (%)
humic acid	81.32 ± 1.05
vermiculite	64.47 ± 1.62
simulated sorbent in soil remediation	92.11 ± 2.26

**Table 2 molecules-26-07478-t002:** Langmuir and Freundlich isotherm parameters for Cr(VI) absorption on the simulated sorbent in soil remediation.

Absorption Isotherm	Parameters		
Langmuir model	*q_m_* (mg/g)	*K_L_* (1/g)	*R* ^2^
	5.57	0.00619	0.996
Freundlich model	*K_f_* (Lmg*^n^*^−1^/g*^n^*)	1/*n*	*R* ^2^
	data 0.553	0.325	0.989

**Table 3 molecules-26-07478-t003:** Extraction efficiencies of Cr(VI) from sorbents into [C_4_mim]Cl.

Extracted Cr(VI) in [C_4_mim]Cl from:	Extraction Efficiency (%)
humic acid	82.85 ± 0.96
vermiculite	21.97 ± 1.11
simulated sorbent in soil remediation	33.48 ± 0.79

## Data Availability

Not applicable.

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
