# Peer review of "Ionic Liquid Extraction Behavior of Cr(VI) Absorbed on Humic Acid–Vermiculite"

_molecules, 2021, doi:10.3390/molecules26247478_

Round 1

Reviewer 1 Report

The manuscript ,,Extraction behavior of Adsorbed Cr(VI) on Humic acid-Vermiculite with an Ionic Liquid,, needs significant improvement and resubmit.

The manuscript is poorly structured, materials and methods are written first, followed by results and discussion.

The whole manuscript is incomprehensible and illegible. Contains unclear results. 

I recommend the authors to care about and improve the manuscript significantly and only then send it to the editorial office.

My decision is to reject this manuscript.

Author Response

A part of description in manuscript has been revised. The structure of manuscript was followed the Microsoft Word template of Molecules.

Reviewer 2 Report

The authors of this manuscript present a chromium extraction method that has the potential to be applied in potential soil decontamination approaches. 

While the work has applied relevance, the authors need to clearly formulate their applied goals in the abstract and the introduction of the manuscript. It is also missing the connection with the real-world contamination situation, limits, and applications. The authors use simulated soil, how does that compare to actual soil structure/texture under field conditions. The manuscript is missing the connection that would extrapolate the relevance of the findings in actual applications. In addition, I believe the authors may also want to address, how Cr toxic levels affect whole ecosystems, document that if studies have been done. Cr is known to be very toxic for plants, not only for animal and human organisms, and is most likely to be the route how it is ingested by animals/humans via the crops that may be exposed to toxic levels of Cr. I would recommend the authors to revisit and expand the "big picture" they want to address through their study and results.

Author Response

As suggestion, the abstract and the introduction sections have been revised.

Abstract section

Cr(VI) can be released into soil as a result of mining, electroplating, and smelting operations. Due to the high toxicity of Cr(VI), its removal is necessary in order to protect ecosystems. Vermiculite is applied in situations where there is a high degree of metal pollution, as it is helpful during the remediation process due to its high cation exchange capacity. The Cr(VI) in vermiculite should be extracted in order to recover it and to reduce its impact on the environment. In this work, adsorption equilibrium data on Cr(VI) in a simulated sorbent in soil remediation (a mixture that includes both humic acid (HA) and vermiculite) fit well with Langmuir isotherm models. The simulated sorbent in soil remediation was a favorable sorbent for Cr(VI) when it was in the tested soil. An ionic liquid, [C4mim]Cl (1-butyl-3-methylimidazolium chloride), was studied to determine its efficiency in extracting Cr(VI) from the Cr-simulated sorbent in soil. At 298 K and within 30 min, approximately 33.48±0.79% of Cr(VI) in the simulated sorbent in soil remediation was extracted into [C4mim]Cl. Using FTIR spectroscopy, the absorbance intensities of the bands at 1032 and 1010 cm-1, which were attributed to the C-O bonds stretching the polysaccharides in the HA, were used to detect the HA changes in the Cr-simulated sorbent in soil remediation before and after extraction. The results showed that Cr(VI) that has been absorbed on HA can be extracted into [C4mim]Cl. Using 1H NMR, it can be observed that the 1-methylimizadole of [C4mim]Cl played an important role in the extraction of Cr(VI), which bonded with HA on vermiculite and was able to transform into the [C4mim]Cl phase.

Introduction section

……

Cr is naturally present in its oxidized state of Cr (III), whereas divalent, tetravalent, and pentavalent Cr are unstable in the environment. Cr(III) is one of the essential elements that is involved in protein metabolism in animals and humans [2]. Cr(VI) is genetically toxic to cells; it has been shown to be a carcinogen, as it affects the functions of deoxyribonucleic acid [3]. It is possible for Cr(VI) to be absorbed into edible plants or vegetables, resulting in them having reduced root and coleoptile growth [4,5]. Moreover, the Cr(VI) concentrations in plants depend on the soluble fraction of it that is present in the soil [6,7]. Once it has entered the food chain, Cr(VI) may cause harm to both animal and human organisms when they ingest these affected edible plants or vegetables. Animal studies by the Institute of Labor, Occupational Safety and Health, Taiwan Ministry of Labor have shown that Cr(VI) can cause malignant tumors, while Cr(III) does not [8]. The United States EPA has classified Cr(VI) as a Group A carcinogen in humans, while Cr(III) is classified as a Group D unclassified substance. Therefore, the removal of Cr(VI) from contaminated soils is essential to avoid its toxic impact on ecosystems. High concentrations of Cr(VI) are often found to bind to humic acid (HA), vermiculite, and kaolin clay [9].  Vermiculite, which has a high cation exchange capacity, is a group of hydrated laminar minerals. Vermiculite is often used as an additive to improve soil structure. Vermiculite is also used as a sorbent in order to absorb the metals that are present in contaminated soil. This results in high concentrations of chromium being accumulated in vermiculite [10,11]. It has also been reported that Cr(VI) can be absorbed on HA. As the pH increases from 1 to 7, the Cr(VI) adsorption to HA decreases from 100% to 34% [12]. Moreover, the removal efficiency of Cr(VI) increases at higher temperatures [13]. In addition, in one study using peanut shells as a sorbent, the Cr(VI) removal ratios increased from 50% to 90% in an aqueous solution when the reaction time increased from one to five hours [14]. Remediation technologies for soil that have been contaminated with metals include biological methods, soil washing, solidification/stabilization, and extraction [15,16,17]. Soil washing is one of the most commonly used techniques that can be applied for the remediation of metal-contaminated soils. Cleaning agents, such as surfactants, are used to remove metals from contaminated soil by extracting these toxins in a liquid phase [18]. Soil washing can also be applied for the recovery of the metals that were used in the sorbents of soil remediation.

Ionic liquids (ILs) have special chemical and physical properties, including high thermal stability, negligible vapor pressure, a broad liquid phase range, and excellent electric conductivity [19]. ILs can be used to replace conventional organic solvents and their impact on the environment is considered to be minimal; therefore, they have also been referred to as green solvents [20]. ILs are made of various ions, thereby possessing the properties of salts. Varying the compositions of anions and cations results in different ILs that can be used for different applications. ILs that remain in a liquid phase at room temperature are called room temperature Ils (RTILs) [21]. Kozonoi et al. utilized 1-butyl-3-methylimidazolium nonafluorobutanesulfonate ([bmi][NfO]) to extract Cs+, Na+, Li+, Sr2+, Ca2+, and La3+ ions from aqueous solutions, with extraction ratios of 5, 24, 39, 79, 81, and 98%, respectively [22]. Metal ions with a higher valence are able to be more easily extracted with ILs. In addition, the extraction efficiencies of metal ions, such as copper, lead, and sodium, are greater when ILs are used than those that are achieved with regular organic solvents, e.g., chloroform [23].

The structure of the Cr(VI) that has been absorbed by the sorbent for remediation in contaminated soil is too complex to reveal the mechanism of absorbance phenomena. In order to understand the effects of Cr(VI) that has been absorbed on vermiculite and that has been extracted with the ionic liquid, a mixture comprising both humic acid (HA) and vermiculite was prepared to simulate the sorbent for remediation in a contaminated soil sample. The adsorption equilibrium data on the Cr(VI) in the simulated sorbent were established. During the extraction process, 1-butyl-3-methylimidazolium chloride ([C4min]Cl) was used to extract Cr(VI) from the Cr-simulated sorbent in soil remediation; the extraction mechanism was explored.

Reviewer 3 Report

Manuscript entitled „Extraction behavior of Adsorbed Cr(VI) on Humic acid-Vermiculite with an Ionic Liquid” submitted to  Molecules is well written and the results are presented in a logical and coherent manner. The paper is adequately organized and the topic is interesting and focuses on the adsorption efficiencies of Cr(VI) with the humic acids, vermiculite, and simulated soil and it raises the current problem. Manuscript is based on unusual in vitro experiment, and the methodology used is correctly selected and varied.

Although the manuscript is well-edited and developed in accordance with the requirements of the Journal, however, small improvements should be introduced that will improve its quality:

  1. References section should be carefully checked and correct according to journal guidelines
  2. In the section Introduction the objective of the study should be added
  3. The mentioned issues should be discussed more.
  4. Please, indicate more the importance of your study (the strengths and weaknesses) and suggest future experiments in the Conclusion part.

Manuscript should be accepted for publication and authors should be prepared to incorporate minor revisions.

Author Response

(The authors gave the same response as above.)

Round 2

Reviewer 1 Report

The authors should improve the structure of the manuscript as I wrote in the previous review. 

They should first describe the materials and methods and then the results and discussion.

The quality of figures 2, 3 and 4 should be improved. It would be appropriate for these figures to look better.

Figure 3: y-axis change ,,(arbitrary units) for (a.u.)

Figure 4 lacks the naming of the y-axis. 

Author Response

Dear Reviewer

Enclosed, you will find the revised manuscript (Ref. No.: molecules-1488344) entitled “Extraction behavior of Absorbed Cr(VI) on Humic acid–Vermiculite with an Ionic Liquid”. The structures of manuscript followed the Microsoft Word template of Molecules. The “results and discussion” section was described after “Introduction” section. All Figures were revised to improve the qualities. Moreover, the manuscript has been reviewed and corrected by an English native scientist from MSPI Author Services. We hope you will find this version of the paper acceptable for publication in a special issue entitled Recent Advances in Green Solvents of Moleculeas.
